# Deep learning-based screening for locomotive syndrome using single-camera walking video: Development and validation study

**Junichi Kushioka**[1,2,3☯], **Satoru Tada**[1,4,5☯]*, **Noriko Takemura**[1,6☯], **Taku Fujimoto**[7], **Hajime Nagahara**[1,8], **Masahiko Onoe**[9], **Keiko Yamada**[10,11], **Rodrigo Navarro-Ramirez**[12], **Takenori Oda**[13], **Hideki Mochizuki**[5], **Ken Nakata**[14], **Seiji Okada**[3], **Yu Moriguchi**[3☯]

1 ayumo Inc., Osaka, Japan, 2 Clinical Translational Science Institute, University of California Los Angeles, Los Angeles, California, United States of America, 3 Department of Orthopaedic Surgery, Osaka University Graduate School of Medicine, Suita, Japan, 4 Department of Clinical Research, National Hospital Organization Osaka Minami Medical Center, Kawachinagano, Japan, 5 Department of Neurology, Osaka University Graduate School of Medicine, Suita, Japan, 6 Department of Artificial Intelligence, Graduate School of Computer Science and Systems Engineering, Kyushu Institute of Technology, Iizuka, Japan, 7 Department of Geriatric and General Medicine, Osaka University Graduate School of Medicine, Suita, Japan, 8 Institute of Datability Science, Osaka University, Suita, Japan, 9 Department of Cardiovascular Surgery, Kishiwada City Hospital, Kishiwada, Japan, 10 Department of Liberal Arts, Faculty of healthcare and welfare, Saitama Prefectural University, Saitama, Japan, 11 Department of Rehabilitation Medicine, The University of Tokyo, Tokyo, Japan, 12 Department of Neurosurgery, Mayo Clinic Florida, Jacksonville, Florida, United States of America, 13 Department of Orthopaedic Surgery, National Hospital Organization, Osaka Minami Medical Center, Kawachinagano, Japan, 14 Department of Health and Sport Sciences, Osaka University Graduate School of Medicine, Suita, Japan

☯ These authors contributed equally to this work.
* tada@neurol.med.osaka-u.ac.jp

**Data Availability Statement:** The datasets used and/or analyzed during the current study, including patients' walking videos, are available upon

## Abstract

Locomotive Syndrome (LS) is defined by decreased walking and standing abilities due to musculoskeletal issues. Early diagnosis is vital as LS can be reversed with appropriate intervention. Although diagnosing LS using standardized charts is straightforward, the labor-intensive and time-consuming nature of the process limits its widespread implementation. To address this, we introduced a Deep Learning (DL)-based computer vision model that employs OpenPose for pose estimation and MS-G3D for spatial-temporal graph analysis. This model objectively assesses gait patterns through single-camera video captures, offering a novel and efficient method for LS prediction and analysis. Our model was trained and validated using a dataset of 186 walking videos, plus 65 additional videos for external validation. The model achieved an average sensitivity of 0.86, demonstrating high effectiveness in identifying individuals with LS. The model's positive predictive value was 0.85, affirming its reliable LS detection, and it reached an overall accuracy rate of 0.77. External validation using an independent dataset confirmed strong generalizability with an Area Under the Curve of 0.75. Although the model accurately diagnosed LS cases, it was less precise in identifying non-LS cases. This study pioneers in diagnosing LS using computer vision technology for pose estimation. Our accessible, non-invasive model serves as a tool that can accurately diagnose the labor-intensive LS tests using only visual assessments, streamlining LS detection and expediting treatment initiation. This significantly improves patient

reasonable request. Participant consent forms specified that only study investigators could access the datasets. Additionally, the videos have elements of personal identification, such as face and gait patterns. Individuals seeking access to the datasets will need the approval of the Ethics Review Board at Kishiwada City Hospital (e-mail: kch@kishiwada-hospital.com, phone: +81-724-45-1000, URL: https://www.kishiwada-hospital.com).

**Funding:** This study was supported by JST SBIR phase 1 (grant number JPMJST2171 to S.T.) and by the 34th research grant by The Nakatomi Foundation (grant number 20211279 to S.T.). The funders had no role in study design, data collection and analysis, decision to publish, or preparation of the manuscript. No authors received a salary from any of our funders.

**Competing interests:** J.K., S.T., N.T., H.N., and Y.M. are co-founders of ayumo Inc., an Osaka University start-up dedicated to the social implementation of artificial intelligence-based walking video analysis. All other authors report no competing interests.

outcomes and marks a crucial advancement in digital health, addressing key challenges in management and care of LS.

## Author summary

Locomotive syndrome (LS) is a condition in which problems with bones, joints, muscles, and nerves cause a decline in the ability to walk and stand. It is estimated that more than 45 million people in Japan have LS. Early detection is vital because LS can be reversed with early treatment. Detecting LS using widely used diagnostic criteria is easy but labor-intensive and time-consuming and, therefore, not widespread enough. To solve this problem, we developed an artificial intelligence model to detect LS by capturing gait videos. Our artificial intelligence model performed as well as or better than orthopedic surgeons in diagnostic accuracy (accuracy: 72% in our artificial intelligence model vs 52% in the average of 6 different orthopedic doctors' clinical diagnosis), but often diagnosed non-LS cases as LS. This non-invasive artificial intelligence model serves as an accurate and simple diagnostic tool for the LS examination, thereby accelerating the timing of behavioral change and treatment intervention. Our model will significantly improve patients' quality of life and enhance the management and care of LS.

## Introduction

Locomotive Syndrome (LS) is defined by decreased walking and standing abilities due to musculoskeletal issues including bones, joints, muscles, and nerves [1]. This decline in musculoskeletal and neurological function significantly impacts daily life activities and independence [2], and the mean prevalence of LS was reported to be 69.8% among the Japanese population [3]. LS is increasingly recognized as a major public health concern due to its impact on reducing physical mobility and function [4]. This condition is prevalent among the elderly and those leading sedentary lifestyles [5,6] and appears earlier in life than frailty [7,8]. When LS progresses, and the decline in physical ability becomes noticeable with symptoms, it is considered physical frailty [7,8]. The stage corresponding to this physical frailty can be described as "LS Stage 3," where the decrease in mobility function hinders social participation [9]. Importantly, the systematic review found that the prevalence of physical frailty is estimated to be 12% in the global population aged over 50 years [10]. Unaddressed, LS can lead to reduced quality of life, higher medical costs, and a greater risk of falls and injuries, placing a significant strain on individuals and healthcare systems worldwide [11].

Management strategies for LS range from pharmacological treatments and surgical interventions for associated musculoskeletal disorders to physical rehabilitation aimed at improving muscle and balance strength [1]. Additionally, addressing symptoms such as pain and numbness, along with correcting nutritional imbalances, forms part of the comprehensive approach to LS treatment [1]. LS is notable for its potential reversibility with appropriate intervention, even conditions associated with late stage of LS may be reversible, underlining the importance of prompt and accurate diagnosis [12]. Although the process for diagnosing LS seemed straightforward by following standardized charts, simpler than using the general Short Physical Performance Battery [13], it requires subjective patient self-report and clinical evaluations by healthcare professionals [14]. This labor-intensive and time-consuming process leads to a gap in routine clinical diagnosis, preventing its wide implementation. Consequently, there

is a growing demand in the medical field for the development of an automated, objective, and cost-effective tool that could improve the efficiency of the LS screening and diagnosis process, thereby mitigating the reliance on manual processes.

Recent progress in deep learning (DL) within the field of computer vision presents new strategies for overcoming diagnostic challenges [15]. Motion analysis, which previously required attaching numerous sensors for full motion capture, has become more convenient through pose estimation models applied to recorded video footage [16]. Innovations in this area have shown that computer vision systems can effectively authenticate individuals based on their walking patterns [17,18]. Moreover, these systems have broadened their utility by estimating age and fatigue levels by analyzing walking videos [19–21]. This technological progress can potentially revolutionize the detection and assessment of human movement disorders, including LS [16].

This study aimed to develop and validate a DL-based computer vision model that identified LS from walking videos recorded with a single camera. By offering an accessible, non-invasive model capable of instantly diagnosing the labor-intensive LS tests through visual assessments alone, we sought to streamline LS detection and accelerate the initiation of treatment.

# Results

## Demographics

Table 1 presents the baseline characteristics of the study participants. In the model creation group, out of 66 participants, 42 (63.6%) were female. The median age for this group was 70 years. In terms of LS classification, the distribution was as follows: 24 participants were identified with stage-3 LS, 9 with stage-2 LS, 15 with stage-1 LS, and 18 were determined to be non-LS. For the external validation group, there were 65 participants, of which 43 (66.2%) were female, with a median age of 69 years. Within this cohort, LS staging was reported as 5 participants with stage-3 LS, 4 with stage-2 LS, 35 with stage-1 LS, and 21 classified as non-LS. Detailed characteristics of the participants in the model creation and external validation groups are shown in the S3 and S4 Appendix, respectively. Across both groups, the predominant age range was 70–79 years old.

**Table 1. Baseline characteristics of the participants.**

|  |  | Model Creation (n = 66) n (%) | External Validation (n = 65) n (%) |
|---|---|---|---|
| Age (y) | <40 | 9 (13.6) | 8 (12.3) |
|  | 40–49 | 7 (10.6) | 5 (7.7) |
|  | 50–59 | 7 (10.6) | 9 (13.8) |
|  | 60–69 | 9 (13.6) | 11 (16.9) |
|  | 70–79 | 20 (30.3) | 25 (38.5) |
|  | > = 80 | 14 (21.2) | 7 (10.8) |
| Gender | Female | 42 (63.6) | 43 (66.2) |
|  | Male | 24 (36.4) | 22 (33.8) |
| LS Stage | 0 n, (%) | 18 (27.3) | 21 (32.3) |
|  | 1 n, (%) | 15 (22.7) | 35 (53.8) |
|  | 2 n, (%) | 9 (13.6) | 4 (6.2) |
|  | 3 n, (%) | 24 (36.4) | 5 (7.7) |

LS; Locomotive Syndrome

## Model Creation and Internal Validation

The data sets for model creation, internal validation, and external validation using a different dataset are described in Fig 1.

In the development and subsequent internal validation of our computer vision model, tailored for LS screening, a strategic emphasis was placed on optimizing the model's sensitivity. This focus is pivotal for a screening instrument intended for the early detection of potential LS cases, ensuring a high degree of accuracy in identifying true positives. The validation process employed a structured cross-validation (CV) methodology, encompassing three distinct segments: CV1, CV2, and CV3, to rigorously assess the model's diagnostic performance. The results are summarized in Table 2.

Our findings reveal notable sensitivity across the CV iterations, with CV1 achieving a sensitivity of 0.81, and both CV2 and CV3 displaying enhanced sensitivity at 0.89. These results yield an average sensitivity of approximately 0.86, illustrating the model's proficiency in accurately detecting LS cases, which is critical for a reliable screening tool. Conversely, specificity scores exhibited considerable variability, with CV1 demonstrating a high specificity of 0.87, contrasted by the reduced specificity observed in CV2 (0.27) and CV3 (0.38), averaging 0.51 across the evaluations. This variability underscores the model's inconsistent ability in identifying true negatives across diverse data sets.

Moreover, the model's Positive Predictive Value (PPV) recorded robust outcomes, with scores of 0.96 (CV1), 0.78 (CV2), and 0.81 (CV3), leading to an aggregate PPV of 0.85. These PPV metrics signify that the model's predictions regarding LS presence are generally precise,

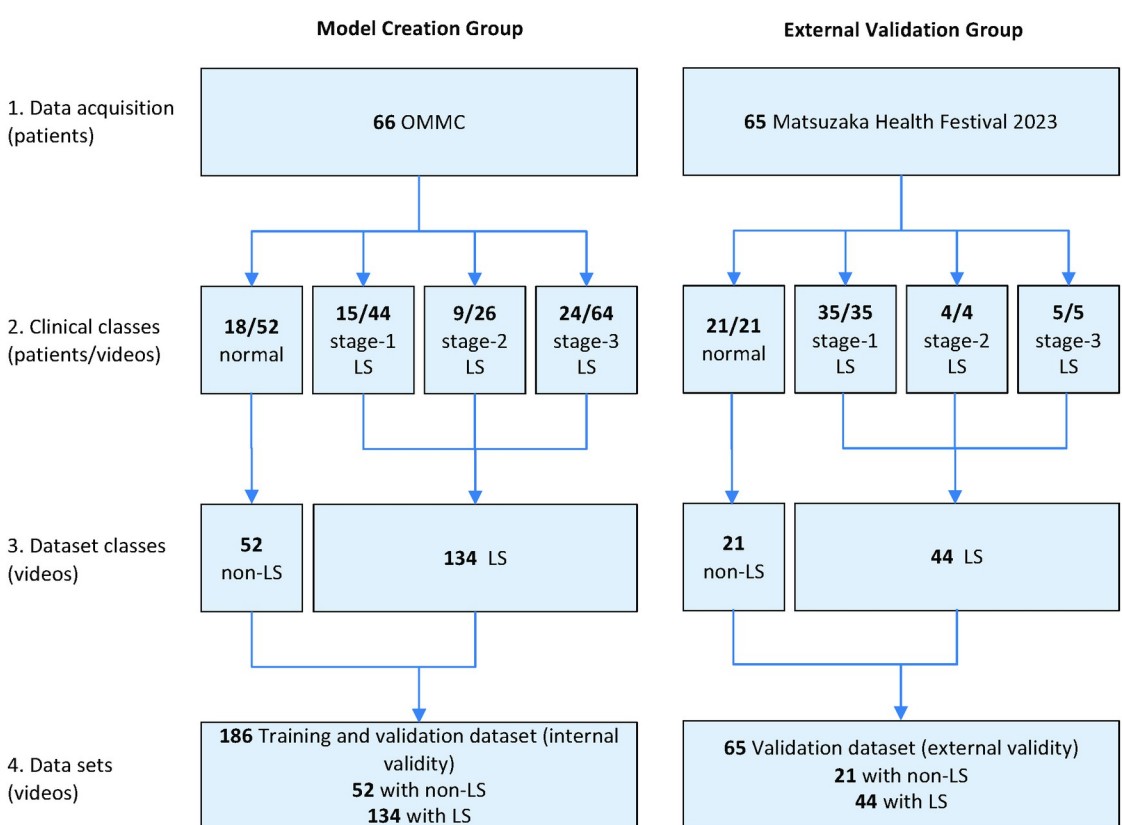

**Fig 1. Data sets for model creation and external validation.**

**Table 2. Internal validation by cross-validation.**

|  | Sensitivity | Specificity | PPV | NPV | Accuracy |
|---|---|---|---|---|---|
| CV1 | 0.81 | 0.87 | 0.96 | 0.56 | 0.82 |
| CV2 | 0.89 | 0.27 | 0.78 | 0.46 | 0.73 |
| CV3 | 0.89 | 0.38 | 0.81 | 0.54 | 0.77 |
| average | 0.86 | 0.51 | 0.85 | 0.52 | 0.77 |

PPV; Positive Predictive Value, NPV; Negative Predictive Value, CV; Cross-Validation

denoting a high level of diagnostic accuracy. However, the Negative Predictive Value (NPV) presented variability and identified areas for improvement, with an average NPV of 0.52 across the CV phases, reflecting the model's fluctuating capability in accurately ruling out non-LS cases.

The accuracy assessments, indicating the model's overall efficacy in correctly classifying LS and non-LS instances, were documented at 0.82 (CV1), 0.73 (CV2), and 0.77 (CV3), with an overall average accuracy of 0.77.

## External Validation

Upon completing the model's development and internal validation, we proceeded with external validation using an independent dataset. This step was crucial for evaluating the model's generalizability and accuracy in a different clinical setting. A Receiver Operating Characteristic (ROC) curve was constructed to provide a detailed assessment of the model's diagnostic performance. The Area Under the Curve (AUC), illustrated in Fig 2, was calculated at 0.75, demonstrating the model's predictive accuracy.

Next, subgroups analysis was conducted to compare patients accurately diagnosed (Accurate) by our developed DL-based model with those inaccurately diagnosed (Inaccurate) (Table 3). The distribution of LS stage differed significantly between the two groups ($p < 0.001$). Further examination of each LS stage revealed that diagnostic accuracy was notably lower for non-LS cases compared to LS stages 1 and above (Fig 3).

Furthermore, we evaluated the diagnostic performance of our DL-based model against the collective judgment of six certified orthopedic surgeons with over ten years of clinical experience. Each doctor independently assessed the same video dataset for the presence of LS in an external validation process. The average diagnostic metrics derived from the doctors' assessments are summarized in Table 4.

Our developed DL-based model exhibited a higher sensitivity (89%) than the average doctors (40%), indicating superior effectiveness in identifying affected patients. However, the doctors demonstrated greater specificity (77% vs 38%), suggesting the DL-based model's higher tendency for false positives. In terms of predictive values, the DL-based model's PPV was slightly lower (75% vs 84%), but its NPV was higher (62% vs 38%) compared to the average of doctors. Overall accuracy favored the DL-based model (72% vs 52%), underscoring its potential to more accurately diagnose LS, despite its limitation in specificity.

## Discussion

This study aimed to develop and validate a DL-based computer vision model for diagnosing LS by analyzing gait patterns in single-camera video recordings using pose estimation (Open-Pose) and graph-structured data through a spatial-temporal graph convolutional network (MS-G3D).

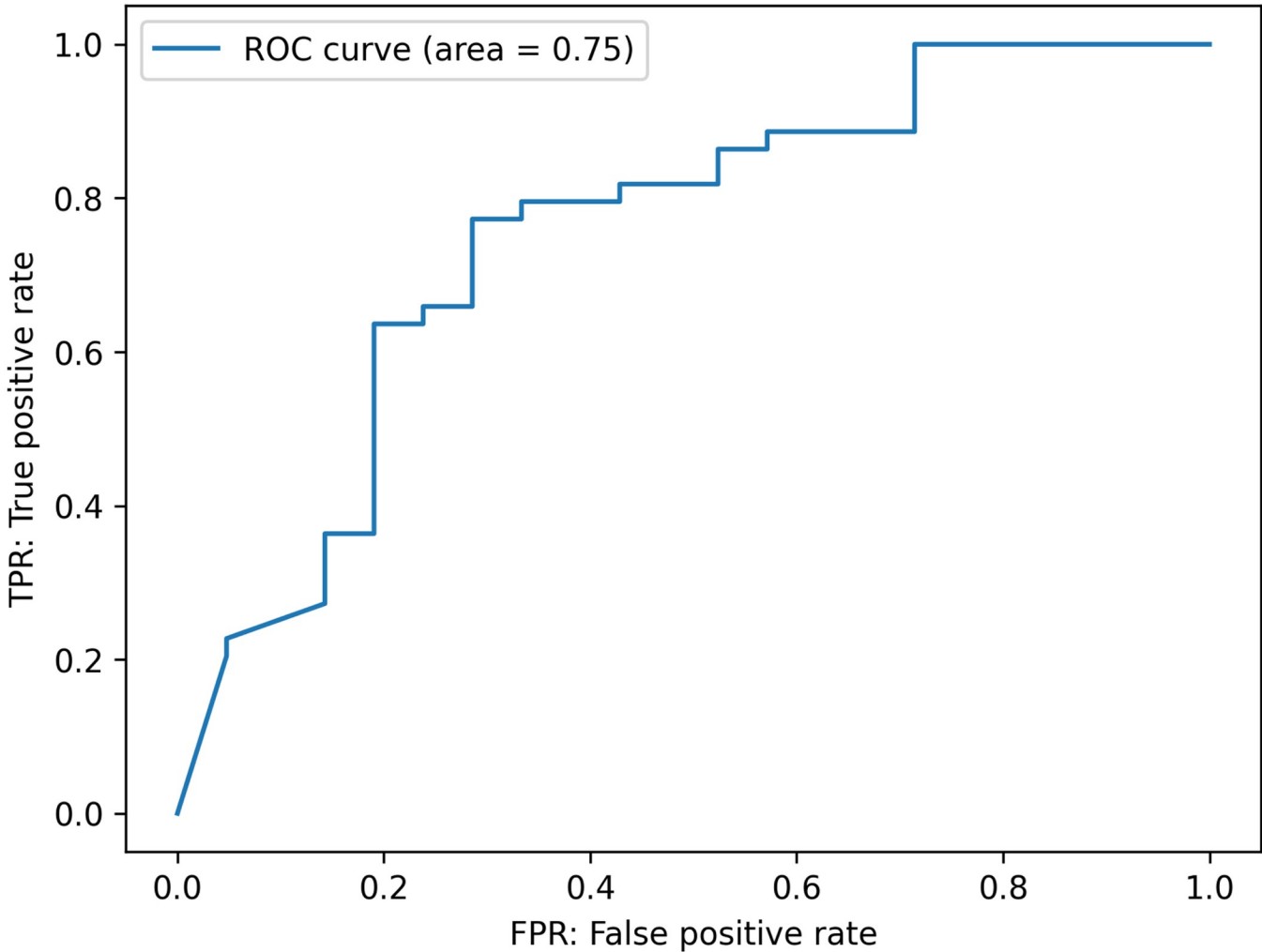

**Fig 2. Area under the curve (AUC) for external validation performance.**

This study introduces the pioneering integration of OpenPose and MS-G3D for the detection of LS, marking a significant advancement in medical diagnostics by combining OpenPose's precise pose estimation from video data with MS-G3D's sophisticated analysis of

**Table 3. Comparison between accurate and inaccurate groups by DL-model for external validation.**

|  |  | Accurate (n = 47) | Inaccurate (n = 18) | P-Value |
|---|---|---|---|---|
| Age (y) | mean, (SD) | 65.8 (15.3) | 57.4 (19.1) | 0.07 |
| Gender | Female n, (%) | 32 (68.1) | 12 (66.7) | 0.99 |
| LS Stage | Stage distribution |  |  | <0.001 |
|  | 0 n, (%) | 8 (17.0) | 13 (72.2) |  |
|  | 1 n, (%) | 32 (68.0) | 3 (16.7) |  |
|  | 2 n, (%) | 3 (6.4) | 1 (5.6) |  |
|  | 3 n, (%) | 4 (8.5) | 1 (5.6) |  |

LS; Locomotive Syndrome

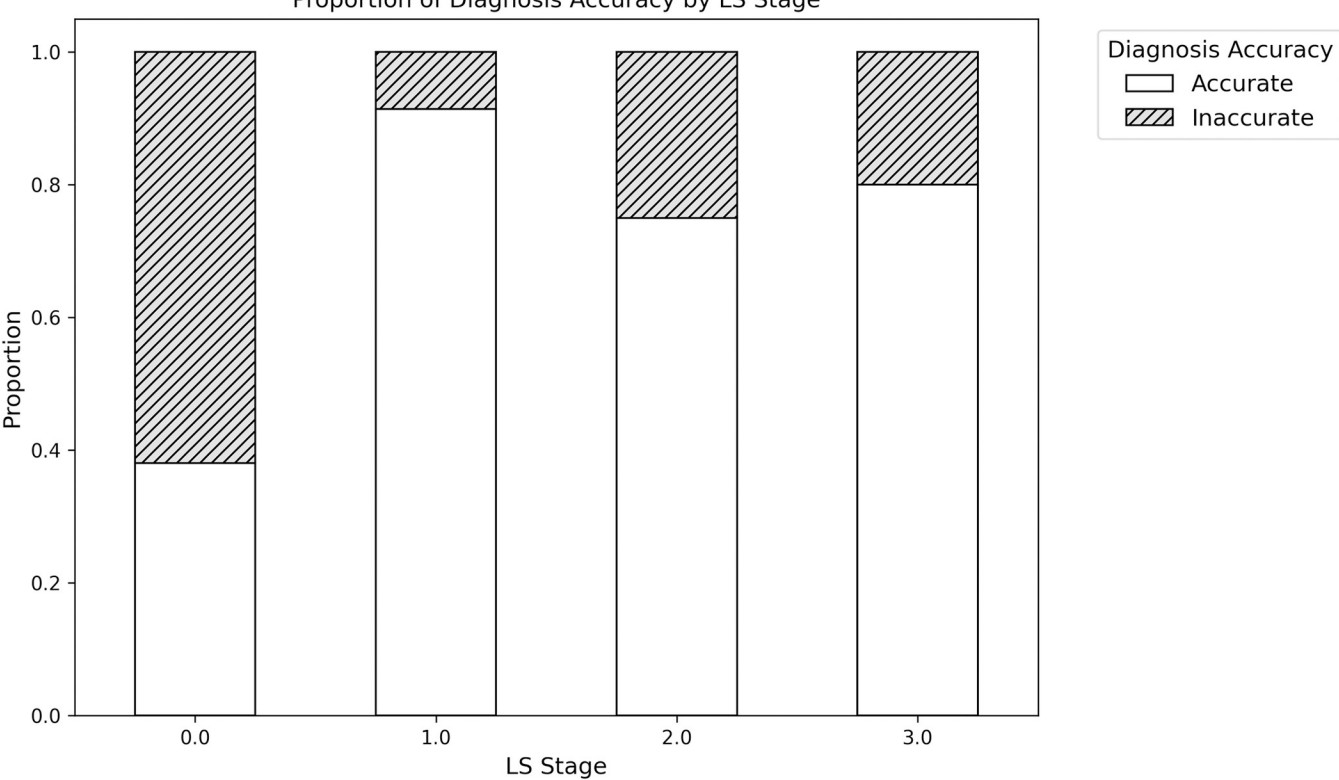

**Fig 3. Proportion of diagnosis accuracy for external validation of DL-based model by LS Stage.**

spatial-temporal graph data. OpenPose's application in our model is supported by its proven effectiveness in various medical contexts, such as analyzing gait abnormalities in individuals with lower limb dysfunction [22] and assessing joint alignment in knee osteoarthritis patients through non-invasive measurement of hip-knee-ankle angles [23], thereby highlighting its utility in clinical medicine and rehabilitation. The incorporation of MS-G3D enhances the model's diagnostic precision by processing the dynamic interactions between body parts during movement, allowing for the detection of subtle gait anomalies indicative of LS. This synergistic use of OpenPose's and MS-G3D's capabilities establishes a comprehensive framework for the early detection and evaluation of LS, leveraging both technologies' strengths to offer an innovative approach to medical condition diagnosis.

The model was subjected to a comprehensive validation process, encompassing both rigorous internal and external validation using an independent dataset, to evaluate its generalizability and diagnostic efficacy across varied clinical environments. During the internal validation, the model exhibited a notable average sensitivity of 0.86, showcasing its ability to accurately identify individuals with LS effectively, which is critical for a screening tool designed for early

**Table 4. The diagnosing performance between DL-based model and doctors' visual examination.**

|  | Sensitivity | Specificity | PPV | NPV | Accuracy |
|---|---|---|---|---|---|
| DL | 0.89 | 0.38 | 0.75 | 0.62 | 0.72 |
| Doctors (n = 6) | 0.40 | 0.77 | 0.84 | 0.38 | 0.52 |

DL; Deep-Learning based model, PPV; Positive Predictive Value, NPV; Negative Predictive Value

detection. Despite achieving a commendable sensitivity, the model's specificity presented less consistency, averaging at 0.51, highlighting a necessity for enhancement in accurately distinguishing individuals without LS. The PPV remained strong at an average of 0.85, indicating that most LS predictions by the model are precise. However, the lower NPV and the fluctuating specificity underscore a possible challenge in reliably excluding non-LS cases. An overall accuracy rate of 0.77 confirmed the model's substantial capability in differentiating LS from non-LS conditions accurately. The development phase's strategic focus on optimizing sensitivity aimed at enhancing the model's application for early LS detection. Although specificity varied, the solid PPV underscores the model's dependable predictive performance, with the synergy between high sensitivity and PPV emphasizing its aptness for early LS screening.

The external validation of our model using an independent dataset yielded an AUC of 0.75, confirming the model's reasonable predictive accuracy in distinguishing individuals with and without LS, thereby verifying its effectiveness as a screening tool in various settings [24]. The consistency of results between internal and external validations highlights the model's stable performance across diverse datasets and clinical environments, essential for its reliability and broader clinical application. Subgroup analysis provided deeper insights into the model's diagnostic accuracy, revealing significant differences in the distribution of LS stages between correctly and incorrectly diagnosed cases, especially noting the model's lower accuracy in identifying non-LS cases. Comparing our DL-based model with the collective judgment of six experienced orthopedic surgeons revealed its high sensitivity (89%) in identifying LS patients, substantially surpassing the doctors' average sensitivity (41%), which is critical for early detection and treatment. However, the model's specificity (38%) was lower than that of the doctors (74%), indicating a tendency towards more false positives. Although the model's PPV was slightly below the doctors' average (75% vs. 83%), its NPV was considerably higher (62% vs. 38%), bolstering its effectiveness in excluding non-LS cases. With an overall accuracy of 72% compared to the doctors' 52%, the model shows significant promise as a diagnostic tool, though the specificity gap underscores the importance of integrating the model with clinical assessment to avoid unnecessary interventions.

Recent advancements in deep learning and computer vision have revolutionized the study of human movement disorders, enabling the automatic tracking and analysis of human movement through video data, thus identifying key body landmarks for the quantitative evaluation of motor functions, which is invaluable for individuals with musculoskeletal and neurological impairments [25]. Markerless Motion Capture (MMC) technology allows for the non-invasive analysis of human motion and has been shown to effectively distinguish between individuals with conditions such as Parkinson's disease and healthy subjects by evaluating symptoms like bradykinesia and tremor [26]. MMC's utility extends to three-dimensional gait assessment in community settings, highlighting its practicality and integration into real-world applications, thus enabling widespread clinical adoption and facilitating patient monitoring, which supports personalized rehabilitation strategies [27]. Moreover, a novel video-based method employing deep learning for the evaluation of bradykinesia in Parkinson's disease, which assesses mobility during daily activities with a focus on fine motor movements like the thumb-index finger distance, indicates enhanced accuracy compared to conventional clinical assessments [28]. This method underscores the potential for advanced remote monitoring and the development of more customized care plans for patients [28].

A recent study leveraging front-view video analysis to automatically classify gait severity in Parkinson's disease (PD) analyzed 456 videos from 19 PD patients and employed a support vector machine to achieve an AUC of 80.88%, highlighting its effectiveness in identifying various gait impairment levels in PD patients and suggesting its applicability in home settings for PD assessments [29]. Similarly, another study developed a neural network model that used

low-dimensional postural data from social interaction videos to identify autism in children with an accuracy of 80.9%, a PPV of 0.784, and a sensitivity of 0.854, analyzing an initial sample of 136 children and an additional test set of 101 children with autism spectrum disorder, utilizing an Long Short Term Memory (LSTM) network to interpret temporal sequences of skeletal key points from short video segments [30]. The promising results of these studies parallel the validation outcomes of our model, which exhibits comparable accuracy, indicating its efficacy for early detection of LS and suggesting its potential for widespread home-based screening and clinical application in monitoring and early detection, similarly facilitating timely interventions for individuals at risk.

Our DL-based computer vision model is the first of its kind, aimed specifically at detecting LS, a condition notable for its potential reversibility with early and appropriate intervention, allowing individuals to regain comfortable mobility [12]. Offering a non-invasive, efficient, and accessible screening method, our model plays a crucial role in the early identification of LS, facilitating faster initiation of treatment. Its introduction marks a substantial advancement in managing LS, providing healthcare professionals with a vital tool for improving patient outcomes and addressing healthcare challenges associated with the condition. Beyond LS, our model has broader implications in digital health by enabling the diagnosis of various gait disorders through walking video analysis, covering conditions like cervical myelopathy, lumbar spinal stenosis, osteoarthritis, Parkinson's disease, cerebrovascular disorders, and peripheral artery diseases. These conditions, which affect gait and require assessments from multiple medical specialties, complicate diagnosis and treatment. Our model's ability to aid in the differential diagnosis of diseases with unique gait patterns empowers general practitioners to efficiently refer patients to specialized care, streamlining the diagnostic process and enhancing patient care.

The primary limitation of our study is its relatively small participant base, potentially weakening the conclusions' strength and generalizability. Additionally, observed low value in the model's specificity and NPV raises concerns about its consistent performance in accurately identifying individuals without LS. The imbalanced training dataset in our current study (non-LS: LS = 52: 134) (Fig 1) could have a potential negative impact on our model, possibly leading to the decreased specificity. Although external validation suggests the model's broad applicability, the utilized dataset's lack of comprehensive demographic representation might limit our findings' universality. Addressing the model's current shortcomings in distinguishing between LS and non-LS cases is crucial. Future initiatives should focus on incorporating a wider array of clinical parameters and training the model with datasets that cover a more extensive range of LS stages and related conditions. Expanding the study to include a larger and more diverse participant group will enhance the research's integrity and the model's relevance to various populations. Conducting further validation studies in diverse clinical and demographic settings is vital to ascertain the model's effectiveness in global healthcare applications, thereby ensuring its contributions to precision medicine are significant and widespread.

Our developed computer vision model represents a significant advancement in the screening of LS, showcasing remarkable sensitivity and predictive accuracy, surpassing the diagnostic capabilities of visual examinations by experienced doctors. This study is at the forefront of employing computer vision technology for pose estimation to diagnose LS, introducing a method that is both accessible and non-invasive. By facilitating early and efficient detection of LS, our model enables quicker commencement of treatment, substantially enhancing patient outcomes. This progress constitutes a pivotal development in digital health, tackling major obstacles in the management and care of LS, and sets a new benchmark for leveraging technology to improve healthcare delivery.

## Methods

### Study design

This study aimed to develop an innovative DL-based computer vision model to identify LS effectively. We developed this model by employing prospectively collected data. To evaluate the effectiveness and broader applicability of our model, we carried out external validation with a separate dataset, also collected prospectively, from an alternative institution.

The study received approval from the Regional Committee for Medical and Health Research Ethics at Osaka Minami Medical Center (OMMC), ensuring compliance with ethical standards and patient safety. Prior to inclusion in the study, all participants provided written informed consent. Additionally, the Ethics Committee of OMMC granted specific approval for this prognostic study (Approval code: R5-42). Consistent with the ethical standards, our study protocol was meticulously designed to align with the principles outlined in the Declaration of Helsinki, guaranteeing respect for the rights and well-being of all participants.

### Participants

The study selected participants based on specific inclusion and exclusion criteria. To be eligible for inclusion, participants were required to meet the following criteria: a minimum age of 20 years, voluntary participation, the ability to independently walk ten meters, consent to undergo the LS risk test, and undergo a medical examination by a certified orthopedic surgeon or neurologist. Individuals were excluded from the study if they were under the age of 20, expressed unwillingness to participate, or were unable to independently walk 10 meters.

For model development, a total of 66 patients who visited the Department of Orthopedics at OMMC between December 22, 2021, and February 21, 2022, were enrolled. Two-thirds of the 66 participants in OMMC were randomly assigned to the model development sample, and one-third were randomly assigned to the internal validation sample. Additionally, for the external validation of the model, 65 participants from the Matsuzaka Health Festival 2023, held in Matsuzaka City on September 10, 2023, were recruited.

### Data collection

For the development of our model, we gathered 186 walking videos from individuals attending the Department of Orthopedics at OMMC. Additionally, for the purpose of external validation, 65 walking videos were collected from attendees of the Matsuzaka Health Festival 2023.

For the video recording, we used a FLIR CHAMELEON3 camera (P/N: CM3-U3-13S2, Edmund Optics Inc., Barrington, USA), with a resolution of 1288x964, 30 FPS, and 1.3 megapixels, and the Edmund Optics UC Fixed Focal Length lens (#33–300) with a 4mm focal length, 12-megapixel C-mount lens with an M61 x 75 filter size, and less than 17.5% distortion.

Participants were instructed to walk down a designated ten-meter path three times. During these walks, each participant was filmed from the right side, with the camera positioned four meters away from the walking path to ensure clear lateral movement capture. The raw footage was saved in MP4 format utilizing Advanced Video Coding (AVC). To specifically analyze stable walking patterns, only the walking sequences occurring between the four to seven-meter marks of the ten-meter path were considered for detailed analysis. For technical reasons, some of the videos recorded could not be played back, and in such cases, no more than two videos per participant were employed as a training dataset.

## LS risk test

The LS risk test comprises three components: a patient-reported outcome measure called the GLFS-25, and two performance tests known as the two-step and stand-up tests. These tests have been previously described in research papers [31]. In brief, the GLFS-25 questionnaire consists of 25 questions, each rated on a Likert scale from 0 to 4, assessing difficulties related to mobility in daily life. Higher scores on this scale indicate a worsening health condition, and the total score, which ranges from 0 to 100, was used for analysis. The content of the GLFS-25 is shown in S1 Appendix. The two-step test involves patients starting from a standing position and taking two successive steps as far as they can. The distance covered by these two steps is divided by the patient's height for standardization. This test is performed twice, and the best result is recorded. The stand-up test is conducted using stools of four different heights (10, 20, 30, and 40 cm). Participants are required to stand up from these stools, either using one or both legs and maintain their posture for 3 seconds after standing. A score between 0 and 8 is assigned based on successful performance, with a higher score indicating better physical condition.

The severity of LS is categorized using LS staging criteria as follows: normal, Stage 1 (the initial stage of decreased mobility defined by specific criteria for the two-step test, stand-up test, and GLFS-25 score), and Stage 2 (an advancing stage of decreased mobility defined by different criteria for the same tests). Additionally, a more severe stage known as Stage 3 (advanced decrease in mobility, limiting social engagement) has recently been defined, with specific criteria for the two-step test, stand-up test, and GLFS-25 score.

## Deep learning-based locomotive syndrome prediction method

Our deep learning-based method for predicting LS is depicted in Fig 4 and encompasses four main steps: video recording, pose estimation, model development, and prediction of LS. The details of each step are described below.

1. Video recording: We recorded subjects walking sideways using a standard digital camera to capture natural gait patterns, ensuring clear visibility of the full body in motion. Forty frames from each video, which typically contain more than one gait cycle and allow for a complete representation of the subject's gait characteristics, were then processed.

2. Pose estimation: The recorded walking videos were processed using the OpenPose framework [32], which provided 2D coordinates for 25 body key points as depicted in Fig 4. The OpenPose simultaneously estimates the heatmap of each joint position and the Part Affinity Fields that represent the relationship between joints by deep learning and estimates the 2D coordinates of each key point using these maps.

3. Model development: The key point data obtained from the previous OpenPose were converted into graph-structured format, representing the body's joints and their connections. We employed a deep learning model, MS-G3D [33], based on spatial-temporal graph convolution networks, to learn the LS prediction model. The MS-G3D model enhances feature extraction by applying convolutions across both spatial and temporal dimensions, leveraging skip connections to encapsulate spatial-temporal relationships effectively.

4. Prediction of LS: The trained model predicts LS by analyzing the subject's gait captured in the video. It outputs a probability score, $P_{LS}$, indicating the likelihood of LS presence. For classification, a threshold of 0.5 is applied; if $P_{LS} \geq 0.5$, the gait is classified as indicative of LS; otherwise, it is classified as non-LS. Sensitivity, specificity, positive predictive value (PPV), negative predictive value (NPV), and accuracy were calculated following the equations shown in S2 Appendix.

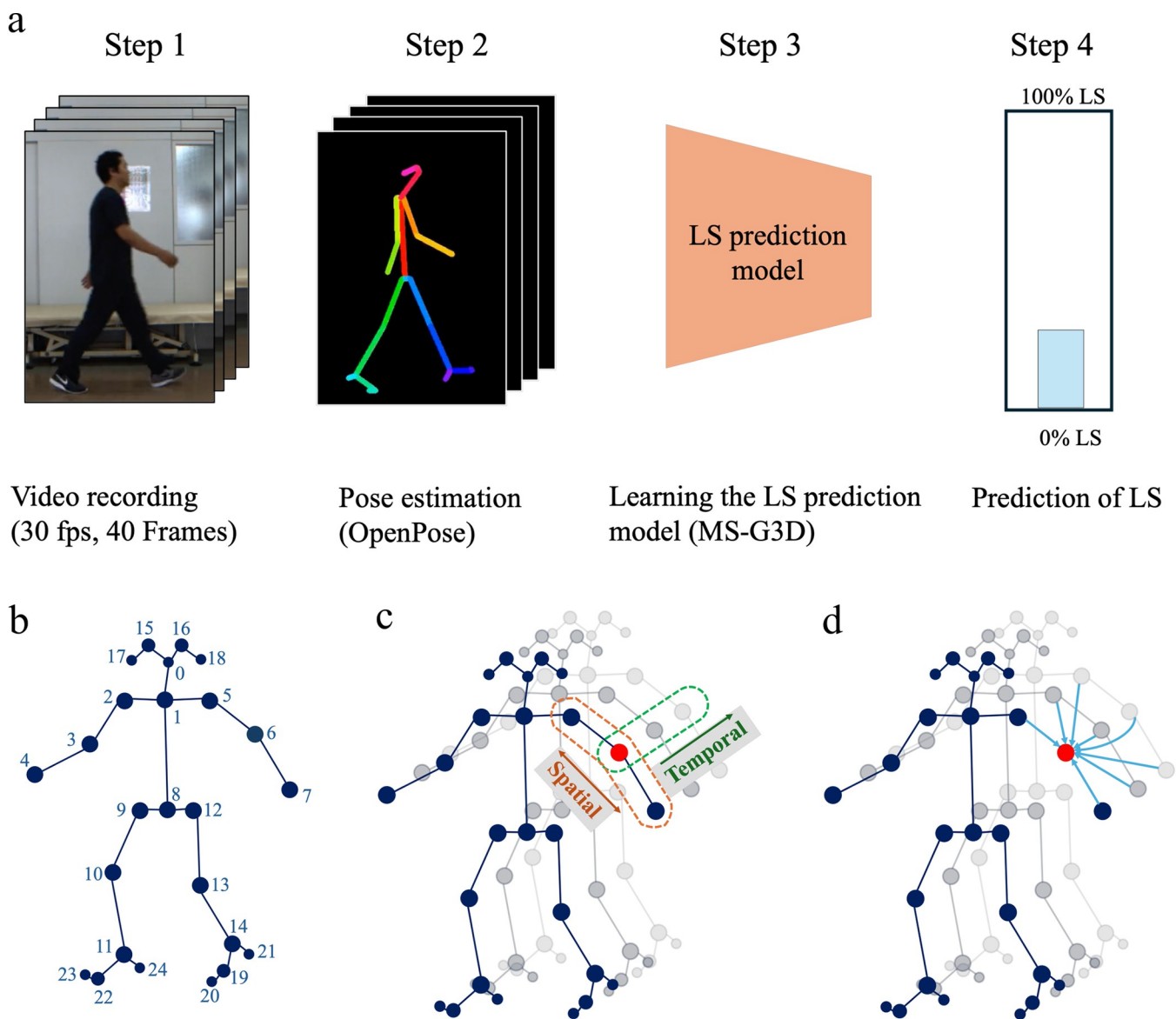

**Fig 4. Deep learning-based locomotive syndrome prediction method.** a. Steps involved in deep learning-based method for locomotive syndrome prediction. Step 1: Video recording of the subject. Step 2: Pose estimation conducted using OpenPose. Step 3: Development of the LS prediction model utilizing MS-G3D. Step 4: Final prediction of Locomotive Syndrome. LS stands for Locomotive Syndrome. b. Skeleton model generated by OpenPose. Depicts the 2D coordinates for 25 key body points as identified by the OpenPose framework. c. Spatial-Temporal GCN-Based LS Prediction Model Utilizing MS-G3D. Diagram of the Spatial/Temporal Graph Convolutional Network (GCN) component. d. Spatial-Temporal GCN-Based LS Prediction Model Utilizing MS-G3D. Enhanced Spatial-Temporal GCN architecture incorporating skip connections.

### Visual diagnosis of doctors for walking videos from external validation dataset

For the external validation dataset, we employed a visual diagnostic approach with six certified orthopedic surgeons, each boasting over a decade of clinical experience. These doctors were asked to evaluate 65 walking video clips, identical to those used in the external validation process. These videos showcased individuals performing a ten-meter walk test, captured from a lateral perspective. The task for each doctor was to determine whether the subjects

demonstrated symptoms of LS. To aid in their assessment, each video was made available for replay up to three times, facilitating a thorough evaluation of potential LS indicators.

## Statistical analysis

Statistical analyses were conducted using Python 3.8, specifically leveraging the SciPy library for our calculations. To evaluate the diagnostic performance of our model, we generated a Receiver Operating Characteristic (ROC) curve and calculated the Area Under the Curve (AUC) as a measure of accuracy during external validation. For numeric variables, differences in means were assessed using independent samples t-tests, allowing us to compare the average values between two groups. For categorical variables, we applied Chi-square tests of independence to determine if there were significant associations between the groups. This comprehensive statistical evaluation was carried out over a period extending from November 11, 2023, to April 3, 2024, ensuring thorough analysis and interpretation of our data.

## Supporting information

**S1 Appendix. The contents of the GLFS-25 questionnaire.**
(DOCX)

**S2 Appendix. The definition of sensitivity, specificity, positive and negative predictive value, and accuracy.**
(DOCX)

**S3 Appendix. Detailed characteristics of the participants in the model creation group.**
(DOCX)

**S4 Appendix. Detailed characteristics of the participants in the external validation group.**
(DOCX)

## Acknowledgments

We would like to express our profound gratitude to Ms. Saeko Doi, Mr. Yasuyoshi Takada, Mr. Yoshiyuki Kuwada, Dr. Miki Tagami, Dr. Takashi Miwa, Dr. Takanori Hazama, and Dr. Ichiro Nakahara for their invaluable contributions to the data collection process in this study.

## Code availability

The underlying code for this study and training/validation datasets is not publicly available for proprietary reasons.

## Author Contributions

**Conceptualization:** Junichi Kushioka, Satoru Tada, Noriko Takemura, Hajime Nagahara, Yu Moriguchi.

**Data curation:** Junichi Kushioka, Satoru Tada, Noriko Takemura, Yu Moriguchi.

**Formal analysis:** Junichi Kushioka, Satoru Tada, Noriko Takemura, Yu Moriguchi.

**Funding acquisition:** Satoru Tada.

**Investigation:** Junichi Kushioka, Satoru Tada, Noriko Takemura, Yu Moriguchi.

**Methodology:** Junichi Kushioka, Satoru Tada, Noriko Takemura, Yu Moriguchi.

**Project administration:** Junichi Kushioka, Satoru Tada, Noriko Takemura, Hajime Nagahara, Yu Moriguchi.

**Resources:** Junichi Kushioka, Satoru Tada, Noriko Takemura, Taku Fujimoto, Hajime Nagahara, Masahiko Onoe, Keiko Yamada, Rodrigo Navarro-Ramirez, Takenori Oda, Hideki Mochizuki, Ken Nakata, Seiji Okada, Yu Moriguchi.

**Software:** Junichi Kushioka, Satoru Tada, Noriko Takemura, Yu Moriguchi.

**Supervision:** Satoru Tada, Noriko Takemura, Taku Fujimoto, Hajime Nagahara, Masahiko Onoe, Keiko Yamada, Rodrigo Navarro-Ramirez, Takenori Oda, Hideki Mochizuki, Ken Nakata, Seiji Okada, Yu Moriguchi.

**Validation:** Junichi Kushioka, Satoru Tada, Noriko Takemura, Taku Fujimoto, Yu Moriguchi.

**Visualization:** Junichi Kushioka, Satoru Tada, Noriko Takemura, Yu Moriguchi.

**Writing – original draft:** Junichi Kushioka, Satoru Tada, Noriko Takemura, Yu Moriguchi.

**Writing – review & editing:** Junichi Kushioka, Satoru Tada, Noriko Takemura, Taku Fujimoto, Hajime Nagahara, Masahiko Onoe, Keiko Yamada, Rodrigo Navarro-Ramirez, Takenori Oda, Hideki Mochizuki, Ken Nakata, Seiji Okada, Yu Moriguchi.

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
