## [Decision Letter · Decision Letter 0]

18 Jun 2024

PDIG-D-24-00163

Deep learning-based screening for locomotive syndrome using single-camera walking video: Development and validation study

PLOS Digital Health

Dear Dr. Tada,

Thank you for submitting your manuscript to PLOS Digital Health. After careful consideration, we feel that it has merit but does not fully meet PLOS Digital Health's publication criteria as it currently stands. Therefore, we invite you to submit a revised version of the manuscript that addresses the points raised during the review process.

Please submit your revised manuscript within 60 days Aug 17 2024 11:59PM. If you will need more time than this to complete your revisions, please reply to this message or contact the journal office at digitalhealth@plos.org. Please include the following items when submitting your revised manuscript:

We look forward to receiving your revised manuscript.

Kind regards,

Ismini Lourentzou

Section Editor

PLOS Digital Health

Journal Requirements:

1. Please provide separate figure files in .tif or .eps format only and remove any figures embedded in your manuscript file. Please also ensure that all files are under our size limit of 10MB.

2. Some material included in your submission may be copyrighted. According to PLOS’s copyright policy, authors who use figures or other material (e.g., graphics, clipart, maps) from another author or copyright holder must demonstrate or obtain permission to publish this material under the Creative Commons Attribution 4.0 International (CC BY 4.0) License used by PLOS journals. Please closely review the details of PLOS’s copyright requirements here: PLOS Licenses and Copyright. If you need to request permissions from a copyright holder, you may use PLOS's Copyright Content Permission form.

Potential Copyright Issues:

Figure 4 includes an image of an identifiable person. Please provide written confirmation or release forms, signed by the subject(s) (or their parent/legally authorized guardian), giving permission to be photographed and to have their images published under our CC-BY 4.0 license. 

Otherwise, we kindly request that you remove the photograph.

3. In the online submission form, you indicated that "The datasets used and/or analyzed during the current study available from the corresponding author on reasonable request". 

3. Uploaded as supplementary information.

Additional Editor Comments (if provided):

Reviewers' comments:

Reviewer's Responses to Questions

**Comments to the Author**

1. Does this manuscript meet PLOS Digital Health’s publication criteria? Is the manuscript technically sound, and do the data support the conclusions? The manuscript must describe methodologically and ethically rigorous research with conclusions that are appropriately drawn based on the data presented.

Reviewer #1: No

Reviewer #2: Yes

Reviewer #3: Partly

2. Has the statistical analysis been performed appropriately and rigorously?

Reviewer #1: No

Reviewer #2: Yes

Reviewer #3: No

3. Have the authors made all data underlying the findings in their manuscript fully available (please refer to the Data Availability Statement at the start of the manuscript PDF file)?

Reviewer #1: Yes

Reviewer #2: Yes

Reviewer #3: No

4. Is the manuscript presented in an intelligible fashion and written in standard English?

Reviewer #1: Yes

Reviewer #2: Yes

Reviewer #3: Yes

5. Review Comments to the Author

Reviewer #1: 1. Prevalence of Locomotor Syndrome:

o The manuscript lacks information on the prevalence of locomotor syndrome, both globally and within the studied population. It is important to provide this context to understand the significance and applicability of your results.

o It is necessary to include references to the global prevalence of locomotor syndrome and, if possible, provide data on the prevalence within the specific population you studied. This information is crucial for interpreting your findings accurately. It is necessary to include the formula to calculate positive and negative predictive values, specificity, etc

2. Equivalence of Training and Validation Groups:

o There is a noticeable imbalance between the training and validation groups in terms of the number of patients and their stages of locomotor syndrome.

o The training group includes 24 patients in stage 3, 9 patients in stage 2, and 15 patients in stage 1. In contrast, the validation group has 5 patients in stage 3, 4 patients in stage 2, and 35 patients in stage 1.

o This discrepancy can affect the validity and reliability of your model’s performance. Please address this imbalance. Possible solutions include:

Expanding the dataset to ensure more balanced groups.

Providing a detailed justification for the current group composition and discussing its implications on the study’s results.

3. Statistical Analysis and Model Validation:

o Given the imbalance in patient distribution, it is important to discuss how this might impact the model’s performance.

o Include statistical analyses that account for the differences in group sizes and stages. Consider performing additional validation using more balanced groups if possible.

o Discuss any potential biases introduced by the current group compositions and how they were mitigated in your analysis

Reviewer #2: The authors implemented a DL-based model for the diagnosis of LS, a disabling but reversible pathology, if recognized early. The paper is well-written and the contents are described completely and clearly. The results, although presenting limitations, are encouraging and represent an interesting first effort in this field.

Reviewer #3: This paper developed a classification model to predict locomotive syndrome (LS) from walking videos. The authors have compared previous studies that have developed similar classification models on different gait pathologies and the results are comparable. 

Minor comments below:

Introduction:

Clear and coherent through the paragraph. Sufficiently elaborated on LS's background and its issues. Appropriately tied up with the study’s aim to develop a model to objectively diagnose LS from walking gait.

Results: 

Line 154: Table 1. The n (%) should be placed in the column header instead to prevent repeating information. Additionally, if the female numbers are already presented, why not present the male numbers as well? This would prevent readers from having to perform their own calculations.

Discussion:

Line 265: How would the “model’s challenge in reliably excluding non-LS cases” affect clinical decisions? 

Line 286: Similar to my comment above, how would the model’s "tendency towards false positives” affect clinical decisions? Would unnecessary resources be put into these false positive cases? How would this affect the current workflow of clinicians to diagnose LS?

Methods:

The statements below are similar and make the paragraph verbose. Please avoid repeating information, especially if they are in the same paragraph.

Line 382: “ …ensuring compliance with ethical standards and patient safety.”

Line 384: “…reinforcing the ethical integrity and transparency of the research process.”

Line 386: “…confirming adherence to ethical guidelines.”

Line 418: “…one to four times”. How did you decide which participant(s) to do one, two, three, or four times? Please elaborate on this.

Line 418: “..from a side view”. Were the video recordings from the left or right side of the participants? From Figure 4, sample images showed were from the right side. Are these consistent for all participants? Please elaborate on this.

Line 413: “…with a resolution of 1288 x 964, 30 FPS,”

Line 421: “at a resolution of 1288 x 964 and a frame rate of 30 FPS”. 

It repeats the information mentioned in line 413. Is it referring to the same camera mentioned in line 413? If yes, I do not see the need to repeat this information in the next paragraph. If it is necessary to mention the use of Advance Video Coding (AVC), would it be possible to combine it with line 413 instead? Or remove the repeating information of resolution and fps in line 421.

Paragraph line 454 on OpenPose. Given that the video recordings were from the left or right side of the participants, did you encounter any issue with occlusions or swapping of lower-limb pose estimation? If yes, what were the steps taken to fix these issues? Were there any checks done before inputting to MS-G3D? 

Line 460: What do you mean by “converted into graph-structured format”? How is this different from the OpenPose default output of pose estimation?

Line 468: "...the gait is classified as indicative of LS; otherwise, it is classified as non-LS." So the model prediction of LS is binary, but in paragraph line 426 on the LS risk test, patients can be classified into multiple stages depending on severity. Please elaborate on why you chose a binary classification instead of replicating the same classification of the LS risk test. Additionally, if the model prediction is binary, how did Figure 3 come about? What does ‘accuracy’ here refer to? I assume is the model vs doctor diagnosis. Then how did you know when the model classified different LS stages here? 

Paragraph line 472: What is the inter-reliability between the 6 orthopedic surgeons? Were all orthopedic surgeons unanimous in classifying LS for all 65 walking video clips?

Line 488: The chi-square test was mentioned but I did not see any chi-square statistical values reported in the results section.

Figure 1. The amount of non-LS and LS cases is imbalanced, almost 2.5 times, in model creation. Do you think this would contribute to the model leaning towards false positives? 

Figure 3, part a. Where did these “40 frames” come from?

6. PLOS authors have the option to publish the peer review history of their article (what does this mean?). If published, this will include your full peer review and any attached files.

**Do you want your identity to be public for this peer review?** For information about this choice, including consent withdrawal, please see our Privacy Policy.

Reviewer #1: No

Reviewer #2: No

Reviewer #3: No

---

## [Decision Letter · Decision Letter 1]

14 Oct 2024

Deep learning-based screening for locomotive syndrome using single-camera walking video: Development and validation study

PDIG-D-24-00163R1

Dear Dr. Tada,

We are pleased to inform you that your manuscript 'Deep learning-based screening for locomotive syndrome using single-camera walking video: Development and validation study' has been provisionally accepted for publication in PLOS Digital Health.

Best regards,

Ismini Lourentzou

Section Editor

PLOS Digital Health

The author response and manuscript revisions have adequately addressed most reviewer concerns. Some concerns remain, particularly regarding the issue of group imbalance and the lack of clarity in distinguishing whether the model is for diagnostic or predictive purposes. The authors have explicitly acknowledge the limitations caused by the imbalance in patient groups in the manuscript. While both limitations could be more clearly stated in the manuscript, this does not detract significantly from the overall value of the study. It is strongly recommended, however, to consider the most recent reviewer comments in the final version of the manuscript, specifically to further clarify the potential nature of the model as either predictive or diagnostic, and include the discussion on why false positives is less of a concern in clinician workflows as well as the interrater reliability scores.

Reviewer Comments (if any, and for reference):

Reviewer's Responses to Questions

**Comments to the Author**

1. If the authors have adequately addressed your comments raised in a previous round of review and you feel that this manuscript is now acceptable for publication, you may indicate that here to bypass the “Comments to the Author” section, enter your conflict of interest statement in the “Confidential to Editor” section, and submit your "Accept" recommendation.

Reviewer #1: (No Response)

Reviewer #3: All comments have been addressed

Reviewer #4: All comments have been addressed

2. Does this manuscript meet PLOS Digital Health’s publication criteria? Is the manuscript technically sound, and do the data support the conclusions? The manuscript must describe methodologically and ethically rigorous research with conclusions that are appropriately drawn based on the data presented.

Reviewer #1: No

Reviewer #3: Yes

Reviewer #4: Yes

3. Has the statistical analysis been performed appropriately and rigorously?

Reviewer #1: No

Reviewer #3: Yes

Reviewer #4: Yes

4. Have the authors made all data underlying the findings in their manuscript fully available (please refer to the Data Availability Statement at the start of the manuscript PDF file)?

Reviewer #1: Yes

Reviewer #3: Yes

Reviewer #4: Yes

5. Is the manuscript presented in an intelligible fashion and written in standard English?

Reviewer #1: Yes

Reviewer #3: Yes

Reviewer #4: Yes

6. Review Comments to the Author

Reviewer #1: The main objective of the study was to develop a model for the prediction and analysis of Locomotive Syndrome (LS). However, the authors neither addressed nor resolved an important issue raised by this reviewer. The issue is that the model was trained and validated using unbalanced groups of patients at different stages of the disease. There was a significant imbalance in the groups used for training and validation, with each group having a different number of patients at each stage of the disease. The authors claim that this was taken into account, and that they are recruiting more patients for a subsequent study. They explain that the model was built using a binary classification: non-LS individuals (control) and LS patients.

The authors argue that the model was trained and validated based on this binary condition—either controls or patients with LS. They state that, in the final analysis, there was an equal proportion of patients (when considered as a single group) in both the training and validation sets. However, they acknowledge an imbalance in the training dataset, with more patients than controls, which led to an increased number of false positives and reduced the specificity of the model. Despite these issues, the authors wish to publish the current version of the predictive model with unbalanced groups. They plan to address this imbalance in a future study, but this is neither ethical nor responsible, as they intend to publish a flawed model in PLOS Data Health.

In another section of their responses to the reviewers, the authors claim to have developed a tool to assist physicians in diagnosing LS (page 9), yet in the revised manuscript (lines 73-75, page 17), they continue to refer to it as a method for prediction and analysis. It appears that the authors may be working on a diagnostic model, which is fundamentally different from a predictive model. This is a significant issue, and the manuscript should not be published in its current form. A predictive model estimates the probability that a disease might occur in the future and how it will progress. To build a reliable predictive model, it is necessary to have balanced groups representing the various stages of the disease as well as a control group—something this study lacks. In contrast, a diagnostic model is designed to determine whether a patient currently has a specific disease.

The authors failed to adequately address the reviewers' concerns regarding the potential bias introduced by the unbalanced groups (pages 8-9). Instead, they simply stated that the training and validation datasets contained equivalent proportions of patients with the disease.

Their response to the question about the tendency for false positives (page 9) is also insufficient. While the authors admit that their model may produce false positives, they dismiss this as a problem by asserting that the final diagnosis will be made by a clinician. However, this overlooks the crucial issue of the imbalance between patients and controls in both the training and validation datasets, as well as the imbalance in disease stages. Therefore, this is not even an appropriate model for diagnostic purposes.

Most of the reviewers’ questions centered on the authors' decision to develop a diagnostic model (despite not specifying this as the aim of the study) rather than a predictive model. The authors did not address these concerns adequately, merely stating that the analysis was based on a binary classification of participants, even though this classification was itself imbalanced

Reviewer #3: Thank you to the authors for adequately addressing my comments.

However, I have 2 additional minor comments.

In response to my earlier feedback, I think it would be good for the authors to include their answers to points 2 and 3 in the manuscript discussion, as it highlighted the author's justification on why the model 'tendency towards false positives' is not a major problem in the clinician’s workflow.

Additionally, I suggest briefly mentioning the interrater reliability score (as mentioned in their answer in point 12) in the paragraph around line 235. Addressing these points would further strengthen the manuscript.

Reviewer #4: The revision is fine.

7. PLOS authors have the option to publish the peer review history of their article (what does this mean?). If published, this will include your full peer review and any attached files.

**Do you want your identity to be public for this peer review?** For information about this choice, including consent withdrawal, please see our Privacy Policy.

Reviewer #1: No

Reviewer #3: No

Reviewer #4: No
